# Biofertilizer with *Bacillus pumilus* TUAT1 Spores Improves Growth, Productivity, and Lodging Resistance in Forage Rice

**Shin-ichiro Agake** [1], **Yoshinari Ohwaki** [2], **Katsuhiro Kojima** [3], **Emon Yoshikawa** [3],
**Maria Daniela Artigas Ramirez** [4], **Sonoko Dorothea Bellingrath-Kimura** [5], **Tetsuya Yamada** [1,6],
**Taiichiro Ookawa** [1,6], **Naoko Ohkama-Ohtsu** [1,6,*] and **Tadashi Yokoyama** [6,7]

[1] Institute of Global Innovation Research, Tokyo University of Agriculture and Technology (TUAT),
3-8-1 Harumi-cho, Fuchu-shi, Tokyo 183-8509, Japan

[2] Central Region Agricultural Research Center, National Agriculture and Food Research Organization (NARO),
2-1-18 Kanondai, Tsukuba, Ibaraki 305-8666, Japan

[3] Faculty of Agriculture, Tokyo University of Agriculture and Technology (TUAT), 3-5-8 Saiwai-cho, Fuchu-shi,
Tokyo 183-8509, Japan

[4] Iriomote Station, Tropical Biosphere Research Center, The University of the Ryukyus, 870 Uehara,
Taketomi-cho, Yaeyama-gun, Okinawa 907-1541, Japan

[5] Institute of Land Use Systems, Leibniz Centre for Agricultural Landscape Research (ZALF),
84 Eberswalder Str., 15374 Müncheberg, Germany

[6] Institute of Agriculture, Tokyo University of Agriculture and Technology (TUAT), 3-5-8 Saiwai-cho, Fuchu-shi,
Tokyo 183-8509, Japan

[7] Faculty of Food and Agricultural Sciences, Fukushima University, 1 Kanayagawa, Fukushima-shi,
Fukushima 960-1296, Japan

\* Correspondence: nohtsu@cc.tuat.ac.jp

**Abstract:** *Bacillus pumilus* strain TUAT1 is a plant growth-promoting bacterium (PGPB) applied as
a biofertilizer, containing its spores, for rice. In this study, we analyzed the short-term effects of
biofertilization on plant growth in the nursery and long-term effects on plant vegetative growth,
yield, and lodging resistance in paddy fields using animal feed rice ('Fukuhibiki' and line LTAT-29
which was recently officially registered as a cultivar 'Monster Nokodai 1') and fodder rice (line
TAT-26). The effects of the biofertilization were analyzed under two nitrogen treatments and at
two transplanting distances in the field. The application of $10^7$ colony forming units (CFU) mL$^{-1}$
bacterial spore solution to seeds on plant box significantly improved the initial growth of rice. The
biofertilizer treatment with this strain at $10^7$ CFU g$^{-1}$ onto seeds in nursery boxes increased the
nitrogen uptake at the early growth of rice in the field, resulting in higher growth at the late vegetative
growth stage (e.g., tiller number and plant height). Furthermore, the improvement of growth led to
increases of not only yield components such as the total panicle number (TPN) and the number of
spikelets in a panicle (NSP) in LTAT-29 but also the straw yield and quality of TAT-26. The lodging
resistances of these forage rice plants were also improved due to the increased root development and
photosynthesis creating tougher culms.

**Keywords:** plant growth-promoting bacteria (PGPB); bacterial endophyte; bacterial spores; *Oryza sativa*;
feed rice; fodder rice; field experiment

## 1. Introduction

Rice is one of the most important crops in the food production system because it is not
only for the human consumption but also can be used as forage for livestock (i.e., feed rice
which is classified as concentrate feed, and fodder rice which is classified as coarse fodder).
Infrastructure, techniques, and knowledge on growing food rice can easily be applied
to feed and fodder rice. The area of forage rice cultivation in Japan has been increasing
since the beginning of the 21-century due to the increased prices of imported other forage

(Figure S1a). However, the domestic self-sufficiency rate of forage has not been increased for 30 years (Figure S1b).

In general, forage rice cultivation requires more nutrients than those of the staple rice because of the higher yielding [1,2]. However, recently, the prices of chemical fertilizers (e.g., nitrogen, phosphorus, and potassium) used in rice farming have become costly owing to exhaustible resources, as reported by Amundson et al. [3]. Moreover, these chemical fertilizers cause water pollution by promoting harmful algal blooms (HABs) and groundwater nitrate contamination [4–7].

In contrast, biofertilizers are environmentally-friendly farming materials that contain beneficial microorganisms, such as plant growth-promoting bacteria (PGPB). Biofertilizers create favorable plant-microbe interactions in the crop rhizosphere, boosting plant growth, improving crop productivity, and reducing the use of chemical fertilizers [8,9]. One of the issues with biofertilizers is that the beneficial effects observed in vitro are not reproduced consistently in agricultural fields due to the complicated interactions in nature masking the effect. Therefore, knowledge of the effectivity of biofertilizers in the field to understand which condition reveals the effect is required for sustainable agriculture. We have researched and developed a biofertilizer, 'Kikuichi/Yume-bio' (Asahi Agria Co., Ltd., Saitama, Japan), containing *Bacillus pumilus* TUAT1 spores as PGPB. *Bacillus pumilus* TUAT1 was isolated from a field in the Tokyo University of Agriculture and Technology, and its entire genome sequence was available, reported by Okazaki et al. [10]. Several genes related to plant growth promotion, including indole-3-acetic acid (IAA) synthesis, siderophore biosynthesis proteins, and acetoin metabolism, have been identified in this strain, and are similar to the mechanisms reported in other plant growth-promoting rhizobacteria (PGPR) [11–21]. Ngo et al. reported that inoculation of *B. pumilus* TUAT1 spores in the human consumption rice cultivar 'Hitomebore', enhanced biomass production more than *B. pumilus* TUAT1 vegetative cells, and this effect was mediated by the enhanced formation of crown roots and lateral roots [22]. Seerat et al. [23] suggested that spores of *Bacillus* species induce root growth by physical contact of spore-specific residues, such as peptidoglycan or polysaccharides, with the root cells. However, the extent of growth promotion by *B. pumilus* TUAT1 differed among the cultivars of the rice core collection in Japan, including 'Hitomebore', suggesting this strain has a selectivity of host plants to exert a higher plant growth-promoting effect [24].

The feed rice (*Oryza sativa* L.) cultivar 'Fukuhibiki' has been developed and adopted as one of the official varieties recommended by the Japanese government since 1993 because of its high grain productivity [25,26]. In addition, the fodder rice varieties for whole crop silage (WCS) such as cultivar 'Tachiaoba', 'Kusahonami', and 'Leaf Star' have also been developed, and the most important factor for the quality of WCS is the high total digestible nutrient yield (TDNY) [27–29]. Controlling rice lodging is one of the goals for maintaining yield and sustainable agricultural cultivation because the biomass and grain yield of crops are greatly affected by lodging [30–32]. A new feed rice line LTAT-29 (officially registered as a cultivar 'Monster Nokodai 1', recently) and a new fodder rice line TAT-26 (Figure S2) [33–36] are characterized by the superior lodging resistance provided by their strong culm, resulting in high yield and biomass production with giant shoots. Appropriate nitrogen fertilization and cultivation density are important factors for high yield and improvement of lodging resistance. Pan et al. [37] showed that optimized nitrogen application (e.g., delayed and reduced) improved the lodging resistance of rice by increasing the accumulation of sclerenchyma cells in the culm wall of lower internodes and upregulation of genes related to lignin and starch synthesis. On the other hand, dense cultivation decreased lodging resistance by increasing the thickness of internodes [38,39].

In a previous study by Win et al., the inoculation of *B. pumilus* TUAT1 in the human consumption rice cultivar 'Koshihikari' was reported to promote root growth in seedlings grown in nursery boxes with different levels of nitrogen fertilization; however, there was no effect on their grain yields in the paddy field [40]. The forage rice cultivars 'Fukuhibiki' and LTAT-29 grown in Wagner pots were also subjected to biofertilizer and rice-husk biochar

treatments by Win et al. [41]. However, the plants treated with biofertilizer alone did not show improvements in grain yield and brown rice yield, but showed enhanced shoot growth at 21 days after inoculation. In another study by the same authors, in addition to biofertilizer and rice-husk biochar treatments, LTAT-29 grown in pots was subjected to two nitrogen fertilization treatments (i.e., basal and split) [42]. Again, there was no enhancement in grain yield due to the biofertilizer, but the shoot biomass was improved with basal nitrogen application. In addition, the shoots of 'Fukuhibiki' at the early growth stage (i.e., two weeks after transplantation) were still significantly different, but those at five weeks after transplantation were not [43]. These studies did not indicate grain yield promotion by biofertilization even though the shoot growth of nursery seedlings and the shoot biomass in the growth stage after transplanting and in harvest were enhanced, suggesting that long-term field management (e.g., effective nitrogen fertilization or transplanting distances for new rice lines) is required for availing the benefits on the yields of biofertilization in the field. The objectives of this study were a) to evaluate the short-term and long-term effects of the biofertilizer, which contains spores of *B. pumilus* TUAT1, on forage rice growths and yields and b) to realize the contribution of the biofertilizer on the quality of fodder rice as WCS and lodging resistance improvements of forage rice in the field.

## 2. Materials and Methods

### 2.1. Bacterial Culture and Biofertilizer Preparation

A single colony of *Bacillus pumilus* TUAT1 obtained by a streak culture using our glycerol stocks was picked from a trypticase soy agar (TSA; Becton, Dickinson and Company, Franklin Lakes, NJ, USA) plate and precultured by overnight shaking in 20 mL of Difco Sporulation Medium (DSM) in a 100 mL Erlenmeyer flask at 30 °C. Then, 15 mL of *B. pumilus* TUAT1 suspension was transferred into 300 mL of DSM in a 1 L Erlenmeyer flask, followed by shaking at 30 °C for 48 h. The culture was centrifuged at $10,000 \times g$ for 30 min, and the cells obtained were washed three times with sterile reverse osmosis (RO) water, and resuspended in 0.8 volume (240 mL) of 0.85% sodium chloride solution. The concentration of the resuspended culture was $4 \times 10^9$ colony forming units (CFU) mL$^{-1}$, counted by plating 10-million-fold serial dilutions on TSA. Therefore, the culture inoculation test suspensions were diluted to $10^1$–$10^8$ CFU mL$^{-1}$ from a $10^9$ CFU mL$^{-1}$ culture. We also developed a biofertilizer named 'Kikuichi/Yume-bio (Asahi Agria Co., Ltd., Toshima City, Japan)' containing $10^7$ CFU g$^{-1}$ of *B. pumilus* TUAT1 spores in ground zeolite material, which is one of the best carriers for biofertilizers, suggested by Hindersah et al. [44].

### 2.2. Seed Preparation

The seeds of *Oryza sativa* L. cultivar 'Fukuhibiki' and lines LTAT-29 and TAT-26 were collected from the fields in Tokyo University of Agriculture and Technology by cultivation. The seeds were sterilized with 70% ethanol for 1 min and soaked in 2500 ppm sodium hypochlorite solution for 15 min. The sterilized seeds were then washed with RO water and then incubated in RO water at 28 °C for 1 d to enhance germination.

### 2.3. Seed Inoculation with Bacterial Culture

After incubation, the seeds were inoculated by soaking in bacterial cultures of different concentrations (0–$10^9$ CFU mL$^{-1}$) for 1 h. Then, four completely infected seeds from each treatment were sown onto plant boxes (0.59 L: $76 \times 76 \times 102$ mm) containing 300 g of Shinano soil (Shinano Baiyoudo Co., Ltd., Nagano, Japan). Each concentration was tested in triplicate for further statistical analyses. The sowed rice seeds were grown in an environment-controlled room at 25 °C, 16 h light (250 µmmol s$^{-1}$ m$^{-2}$), and 8 h dark until the seedlings reached the soil surface. Subsequently, they were moved to a greenhouse for further growth, and harvested 18 days after sowing for measuring shoot and fresh root weights.

*2.4. Nursery Preparation and Rice Transplantation*

A total of 120 g of seeds of each 'Fukuhibiki', LTAT-29, and TAT-26 were sown in a nursery box (4.5 L: 280 × 580 × 28 mm) with Shinano soil, and the seeds were sandwiched between 3 kg of nursery bed soil and 1 kg of cover soil. For the inoculation treatment, the nursery soil was mixed with 'Kikuichi /Yume-bio' as the bacterial source; 5% ($w/w$) biofertilizer, i.e., 200 g for 4 kg of nursery soil, was mixed with only the cover soil. The uninoculated treatments were not mixed with biofertilizer. The 'Fukuhibiki' seedlings were grown in the plastic greenhouse, where we generally cultivate rice nurseries, for 21 days, and the LTAT-29 and TAT-26 seedlings were grown for 14 days. Ten seedlings were randomly collected, and the soil from their roots was removed carefully using tap water. The seedling collection was conducted in triplicate. Shoot length, fresh weight, and leaf number of the harvested seedlings were analyzed. The ratio of shoot fresh weight to shoot length (RSWL) was then calculated. Root basements (one gram) were also collected for plate counting on tryptic soy agar with 100 mg $L^{-1}$ rifampicin and streptomycin to confirm that *B. pumilus* TUAT1 colonized the roots. Roots were mashed in a mortar with 9 mL of RO water. The root suspension was heated at 65 °C for 1 h to kill vegetative cells for spore counting, whereas it was not applied to total number (vegetative cells + spores) counting. The number of vegetative cells was calculated by subtracting the number of spores from the total number counting.

The nursery seedlings were transplanted into a 2025 $m^2$ (27 × 75 m) paddy field located in Harimichi, Nihonmatsu City, Fukushima Prefecture, Japan (37°36′11.4″ N, 140°34′52.6″ E), managed by a professional rice farmer. The soil of the field was sampled before fertilization to analyze their fertility. Soil samples were collected from three areas in a field; the soils were gathered from five points following the diagonal sampling method in each area [45]. The soil was classified as gray lowland soil, and its characteristics (Table S1) were analyzed by Farming Technologies Japan Co., Ltd., Tokyo, Japan. The paddy field was fertilized with 0.7 kg $m^{-2}$ of organic compost for yearly soil improvement and 18 g $m^{-2}$ of $K_2O$ in potassium chloride fertilizer (ITOCHU Corporation, Tokyo, Japan) to reduce radioactive Cs absorption by plants. For the experimental treatment (Table 1), we included an application of chemical nitrogen fertilization in two different amounts as follows: 2 g $m^{-2}$ of N in the form of ammonium sulfate fertilizer (henceforth referred to as N2; Nippon Steel Corporation, Tokyo, Japan) and 4 g $m^{-2}$ of N as a mixture of ammonium sulfate fertilizer and release-controlled urea fertilizer (henceforth referred to as N4; LP coat sigmoid type released in 120 days; LPS120, JCAM Agri Co. Ltd., Tokyo, Japan), mixed in a 1:1 ratio. The fields were equally basally fertilized with calcium superphosphate (Katakura and Co-op Agri Corporation, Tokyo, Japan; 6 g $m^{-2}$ calculated as $P_2O_5$) and potassium chloride (ITOCHU Corporation, Tokyo, Japan; 6 g $m^{-2}$ calculated as $K_2O$). We also used two transplanting distances: 15 cm internal length and 30 cm width (i.e., 22.2 hills $m^{-2}$), which is the conventional distance, and 30 cm of internal length and 30 cm width (i.e., 11.1 hills $m^{-2}$), which is generally sparse transplanting in Japan (Table 1).

**Table 1.** The field design of the experiment.

| Distance (Length by Width, cm) | Chemical N Fertilization | Biofertilizer |
|:---:|:---:|:---:|
| 15 × 30 | N2 | Control |
| 15 × 30 | N2 | BF |
| 15 × 30 | N4 | Control |
| 15 × 30 | N4 | BF |
| 30 × 30 | N2 | Control |
| 30 × 30 | N2 | BF |
| 30 × 30 | N4 | Control |
| 30 × 30 | N4 | BF |

N2: 2 g $m^{-2}$ of N, which forms ammonium sulfate fertilizer. N4: 4 g $m^{-2}$ N, which is a mixture of ammonium sulfate and release-controlled urea fertilizer, mixed at a ratio of 1:1. BF: Biofertilization treatment.

### 2.5. Plant Growth Analysis

The plant height, number of tillers, and soil plant analysis development (SPAD) values were measured as growth parameters at 8 and 13 weeks after transplanting (Table 2). Three plots were used for each treatment (Table 1). Ten tillers per plot were manually counted and calculated for a square meter. Ten plant heights in each plot were randomly measured using a ruler. The SPAD value is a developed technology for the measurement of in situ chlorophyll content [46–48]; SPAD value of the second leaf from the uppermost fully expanded leaf was averaged from the measurement of three points using SPAD-502Plus (Konica Minolta Inc., Tokyo, Japan), and we randomly scanned five rice plants per plot.

**Table 2.** The periods of surveys and the growing stages of each rice variety.

| DAT (DAS) | Trans-Planting | 1st Survey (8 Weeks) | 2nd Survey (13 Weeks) | Heading Stage | Lodging Measurement | Harvest |
|---|---|---|---|---|---|---|
| 'Fukuhibiki' | 0 (21) | 60 (81) | 92 (113) | 86 (107) | 119 (140) | 137 (158) |
| LTAT-29 | 0 (14) | 61 (75) | 91 (105) | 103 (117) | 137 (151) | 147 (161) |
| TAT-26 | 0 (14) | 61 (75) | 92 (106) | 110 (124) | 147 (161) | 137 (151) |

DAT: days after transplantation into the field. DAS: days after sowing in the nursery box.

### 2.6. Yield Analysis

Matured 'Fukuhibiki' and LTAT-29 plants were harvested for feed yield analysis at 137 and 147 days after transplanting, respectively (Table 2). One harvest plot had 28 rice plants in the field, and samples were collected from triplicate areas. The harvested rice plants were dried for two weeks in a greenhouse, and naturally dried rice plants were threshed for further analysis after counting the number of panicles. The total number of threshed rice seeds was counted using a Multi Auto Counter (Fujiwara Seisakusho Co. Ltd., Tokyo, Japan), followed by the removal of immature rice seeds using the wind selection method, and the hulls of the selected rice seeds were taken out at the same time using a Test Rice Huller (Satake Corporation, Hiroshima, Japan). The Multi Auto Counter was used to detect the number of hulled brown rice seeds, and the gross brown rice yields were weighed. The number of spikelets in a panicle (NSP), percentage of ripened rice seeds to the total number of rice seeds (PRR), weight of 1000 gross brown rice seeds (WBR), total panicle number in a square meter (TPN), and GBRY in 100 square meters were determined. The correlations between the yield components of inoculated and uninoculated plants were analyzed using scatter plots. The slopes of the approximation straight lines, coefficients of determination ($R^2$), and $p$ values of the correlations were calculated.

Twelve rice plants of TAT-26 were collected from one plot at 20 weeks after transplanting, i.e., at the yellow ripening stage, which is the ideal timing for WCS [28]. The rice was harvested from three areas for each treatment for statistical replicates. The TPN and the dry weight of straws and panicles in a square meter were measured. The N concentration in the shoots was analyzed using an NC analyzer (SUMIGRAPH NCH-22F, Sumika Chemical Analysis Service Ltd., Osaka, Japan). The parameters for analyzing the quality of WCS are as follows: crude protein (CP), crude fat (CFA), crude fiber (CFI), crude ash (CA), nitrogen-free extract (NFE), non-fiber carbohydrate (NFC), acid detergent fiber (ADF), neutral detergent fiber (NDF), organic cell contents (OCC), organic cell wall (OCW), and organic a fraction in OCW (Oa), which is rapidly hydrolyzed by cellulase, and organic b fraction in OCW (Ob), which is not digested by cellulase. These values were analyzed by Snow Brand Seed Co. Ltd., Hokkaido, Japan. The total digestible nutrients (TDN) were calculated using the following equation: TDN = 16.651 + 1.494 × (OCC + Oa) − 0.012 (OCC + Oa)$^2$ [49,50]. The total biomass of the shoot, the ratio of panicle biomass to total shoot biomass, the nitrogen uptake of the shoot, and the TDNY were also calculated.

### 2.7. Evaluation of Plant Lodging Resistance

The measurement for calculating lodging factors (e.g., bending moment and lodging index) was in accordance with previous reports [51,52]. The plant height, shoot fresh

weight, the number of culms, and the pushing resistance 15 cm above the ground surface of 10 rice plants were randomly measured in the field to calculate the factors. The growth stage was 27 d after the heading stage (Table 2). The pushing resistance was measured using a digital force gauge (IMADA Co. Ltd., Aichi, Japan). The calculation formulas are as follows:

Equation (1). The formula for Bending moment

$$Bending\ moment\ (\text{kg·cm}) = shoot\ length\ (\text{cm}) \times shoot\ fresh\ weight\ (\text{kg}) \tag{1}$$

Equation (2). The formula for Lodging index

$$Lodging\ index\ (\%) = \frac{Bending\ moment\ (\text{kg·cm})}{Pushing\ resistance\ \left(\text{kg hill}^{-1}\right) \times 15\ (\text{cm})} \times 100 \tag{2}$$

*2.8. Statistical Analysis*

All data collected in this experiment for multiple comparisons, such as multi-way analysis of variance (ANOVA) and Dunnett's tests, were analyzed using SPSS ver. 23 (IBM SPSS Statistics, Armonk, NY, USA). Dunnett's test was selected for a multiple comparison with a single control. Student's *t*-test for two-sample comparisons and Pearson's correlation coefficient tests were performed using Microsoft Excel 365 (Microsoft Corporation, Redmond, WA, USA).

**3. Results**

*3.1. Inoculation of Bacillus pumilus TUAT1 Spores Suspension to the Forage Rice in the Plant Box*

All healthy rice plants were harvested from each triplicate box 18 days after sowing, and fresh shoot and root weights were measured (Figure S3). Analysis of variance (ANOVA) of the results showed that the fresh weights of roots were significantly different among the cultivars ($p = 0.005$), whereas the shoot weights were not significantly different at 18 days after sowing (Table S2). The concentration of inoculants significantly affected the root growth but did not affect the shoot growth. The most effective concentration of the inoculant was $10^7$ CFU mL$^{-1}$, calculated using Dunnett's test; the *p* value was less than 0.001 (Table S3). The ratio of the fresh weight of inoculated plants to that of the non-inoculated plants of each cultivar is depicted using a bar graph (Figure 1). Based on Dunnett's test for each rice variety, the roots of 'Fukuhibiki' and TAT-26 were significantly promoted when inoculated with $10^7$ CFU mL$^{-1}$ bacterial suspension ($p = 0.021$ and 0.048, respectively). However, LTAT-29 did not even show a 35% increase in plants inoculated with $10^7$ CFU mL$^{-1}$ bacterial suspension, compared to the non-inoculated plants.

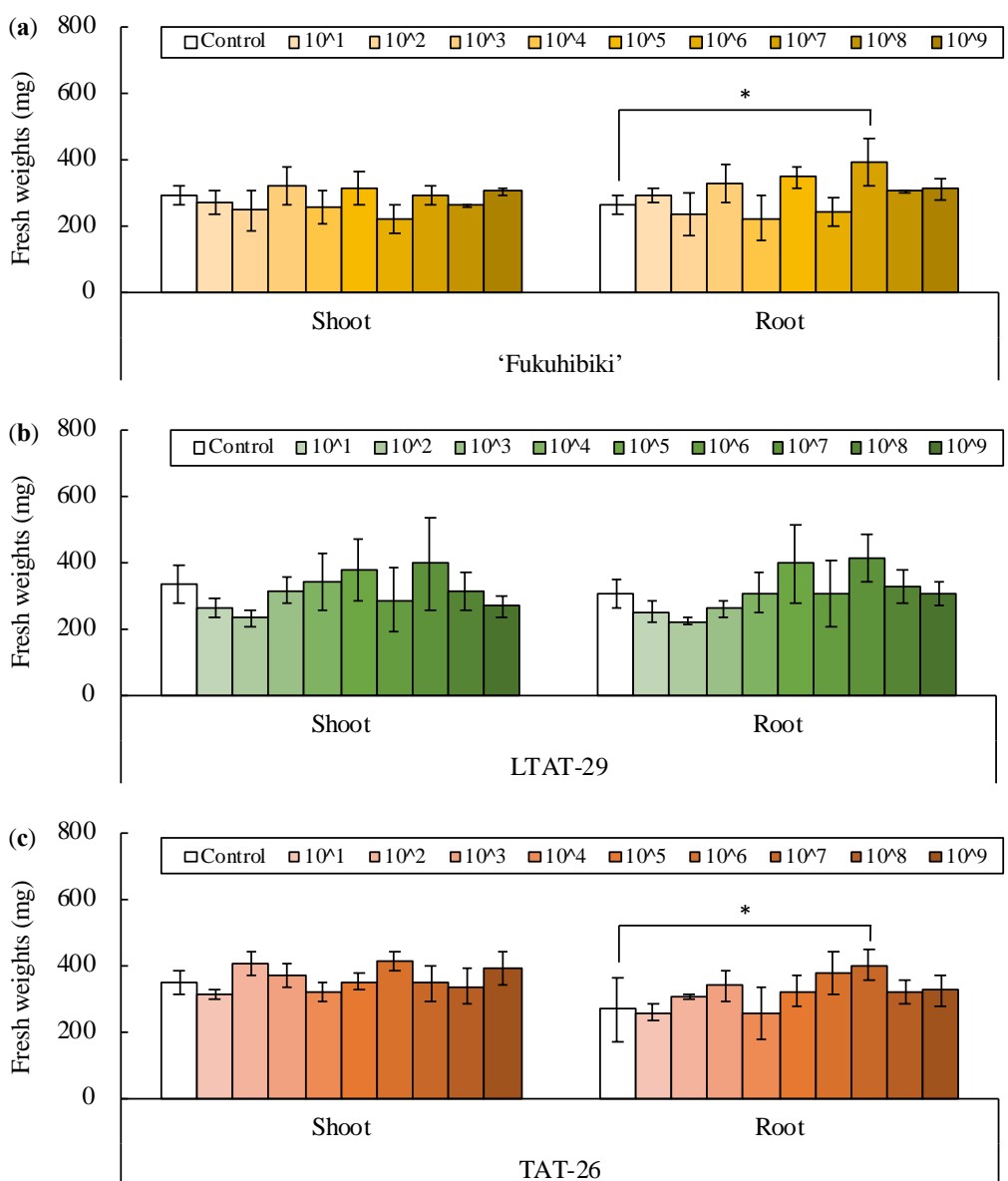

**Figure 1.** The fresh weights of (**a**) 'Fukuhibiki', (**b**) LTAT-29 and (**c**) TAT-26 inoculated with each concentration of *B. pumilus* TUAT1 spores cultivating in plant boxes for 18 days. * indicate significant differences at $p \leq 0.05$ levels between control and treatment, respectively (Dunnett's test, two-sided). Error bars indicate standard deviation (SD). $n = 3$.

### 3.2. The Effects of the Biofertilizer on the Nursery Seedlings of Forage Rice

The biofertilizer Kikuichi/Yume-bio containing *B. pumilus* TUAT1 spores at $10^7$ CFU g$^{-1}$ in the zeolite pores enhanced the shoot growth of rice cultivated in the nursery box for 14 days in LTAT-29 and TAT-26, and for 21 days in 'Fukuhibiki' (Figure 2 and Table S4). The shoot lengths of 'Fukuhibiki', LTAT-29, and TAT-26 were increased by 45%, 37%, and 40%, respectively, compared to that of the control (Figure 2d). The fresh shoot weight of 'Fukuhibiki' and LTAT-29 increased by 41% and 17%, respectively, whereas that of TAT-26 did not show any increase compared to the control (Figure 2e). In contrast, the fresh root weight of 'Fukuhibiki' was decreased and that of LTAT-29 and TAT-26 did not show any significant increase compared to the control. The number of *B. pumilus* TUAT1 that colonized 'Fukuhibiki', LTAT-29, and TAT-26 during the tranplant-ing period were $5.4 \times 10^5$, $3.1 \times 10^5$, and $1.7 \times 10^5$ CFU g$^{-1}$ of the fresh root weight, respectively (Table S5).

The number of colonized vegetative cells in each treatment group was higher than the number of spores.

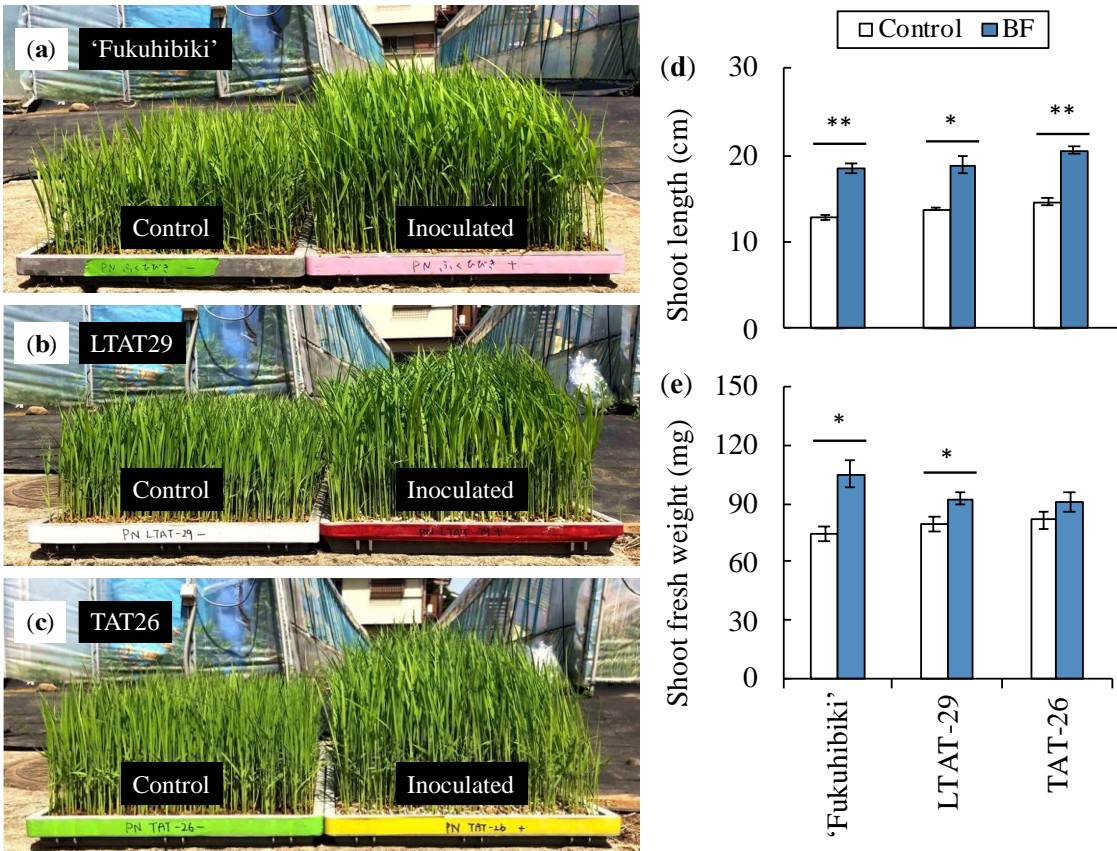

**Figure 2.** The shoot inoculated with Kikuichi/Yume-bio in the nursery box at the transplanting stage. The seedlings of (**a**) 'Fukuhibiki' at 21 days after sowing, and (**b**) LTAT-29 and (**c**) TAT-26 at 14 days after sowing on the nursery box are shown. The measured (**d**) shoot length and (**e**) shoot fresh weight are also indicated. * and ** indicate significant differences at $p \leq 0.05$ and 0.01 levels between control and biofertilizer application in each treatment, respectively (*t*-test, two-sided). Error bar indicates standard deviation (SD). *n* = 3.

### 3.3. The Effects of the Biofertilizer on Tillers Number, Plant Height, and Nitrogen Concentration in the Vegetative Growth Stage of Forage Rice

The results of the field surveys are summarized in Figure 3, Tables S6 and S7. At the early vegetative growth stage of forage rice (i.e., 60 and 61 days after transplanting), the biofertilizer affected the SPAD values, which is the index of nitrogen accumulation in the leaf, while it did not affect tiller number and plant height. There were considerable differences in the tiller number, plant height, and SPAD values among the varieties. The effects of cultivation distance on tiller number and SPAD value are shown in Table S6. Furthermore, differences in fertilization affected plant height and SPAD value. Biofertilizer inoculation significantly enhanced the tiller numbers in 'Fukuhibiki' with 15 × 30 cm and N4 treatments, and the SPAD value of LTAT-29 with 30 × 30 cm and N2 treatments. However, biofertilizer inoculation did not increase plant height in any variety. Moreover, 'Fukuhibiki' was statistically suppressed in the 30 × 30 cm and N4 treatments. In TAT-26, no improvement was observed in plant height, tiller number, or SPAD value.

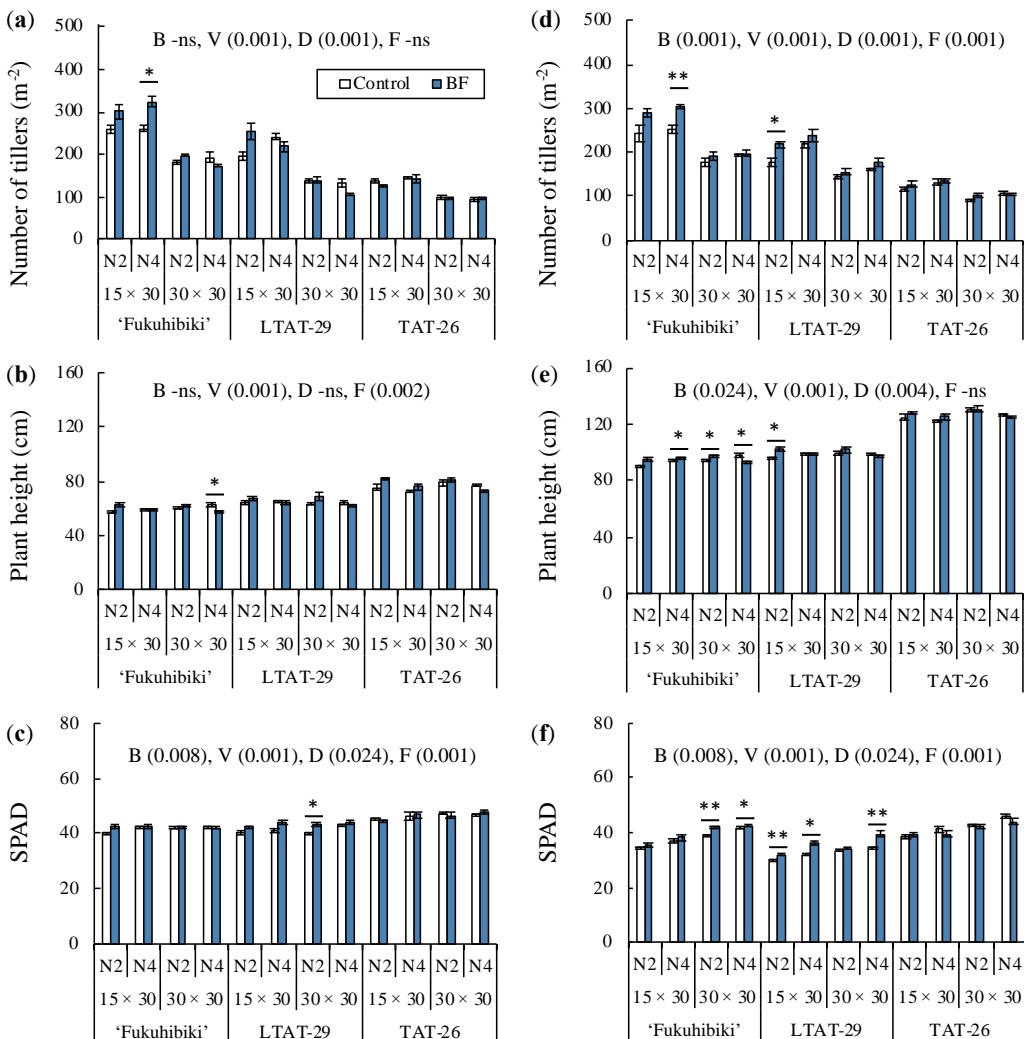

**Figure 3.** Summary of surveys on the paddy field. (**a**) The number of tillers, (**b**) the plant height, and (**c**) SPAD value measured at 8 weeks after transplanting. (**d**) The number of tillers, (**e**) the plant height, and (**f**) SPAD value measured at 13 weeks after transplanting. B: Biofertilizer, V: Variety, D: Distance, F: Fertilization. Each number inside of parenthesis is *p* value (ANOVA). * and ** indicate significant differences at $p \leq 0.05$ and 0.01 levels between control and biofertilizer application in each treatment, respectively (*t*-test, two-sided). Error bar indicates standard deviation (SD). *n* = 3.

At the late vegetative growth stage of rice (i.e., 91 and 92 days after transplanting), most of the treatments, such as biofertilizer, variety, distance, and fertilization, affected the tiller number, plant height, and SPAD value, whereas the fertilization did not affect plant height (Table S7). In 'Fukuhibiki' with 15 × 30 cm and N4 treatments, biofertilizer inoculation considerably improved tiller number and plant height, while in 'Fukuhibiki' with 30 × 30 cm and N2 treatments, it increased the plant height and SPAD value. However, in 'Fukuhibiki' with 30 × 30 cm and N4 treatments, biofertilizer inoculation enhanced the SPAD value but inhibited plant height. No significant difference in tiller number, plant height, and SPAD value was observed compared between inoculated and non-inoculated in the 15 × 30 cm and N2 treatments in 'Fukuhibiki'. In contrast, biofertilizer inoculation increased tiller number, plant height, and SPAD value in LTAT-29 with 15 × 30 cm and N2 treatments. Moreover, the SPAD value improved at both 15 × 30 cm and 30 × 30 cm under N4 treatment. In our study, biofertilizer inoculation did not enhance the growth of TAT-26 in the late vegetative stage.

Biofertilizer inoculation and transplanting distances showed interactive effects on tiller numbers during both early and late vegetative growth stages (Tables S6 and S7). Both

interactive effect of biofertilizer with distance or fertilization was observed on plant height at early and late stages. Furthermore, the SPAD values of those stages were affected by the interactive effects of biofertilization and variety.

### 3.4. The Effects of the Biofertilizer on the Feed Rice Yields

The analyzed yield components are listed in Table 3. Biofertilizer treatment significantly improved the TPN and GBRY. All yield components were affected by the rice variety and transplanting distance. Fertilization treatment affected TPN, although it did not affect the yield. The effects of biofertilization (BF) and no biofertilization (control) are also displayed in Table 3. In 'Fukuhibiki', the TPN was improved in response to BF under all treatments. However, the NSP was decreased by BF in 'Fukuhibiki' with 30 × 30 cm and N2 treatments. Furthermore, BF increased the WBR in 'Fukuhibiki' with 30 × 30 cm and N4 treatments. The yield of inoculated 'Fukuhibiki' with 15 × 30 cm and N2 treatments was significantly higher compared with the control ($p = 0.0024$). There was no statistical difference in the other treatments of 'Fukuhibiki'; however, BF improved the yield compared with the control. In 'Fukuhibiki', the highest yield was obtained from plants subjected to BF, 15 × 30 cm, and N2 or N4 treatments. The TPN of LTAT-29 inoculated with biofertilizer, planted at 15 × 30 cm spacing and subjected to N2 treatment was higher than that without inoculation, resulting in significant yield improvement. Thus, the highest yield in LTAT-29 was in plants subjected to BF, 15 × 30 cm, and N2 treatments. Although the TPN improvement by BF was also observed in the treatment with 30 × 30 cm and N2, it did not lead to statistically higher yield. The interactive effect of the biofertilizer with variety or distance was evident on both TNP and GBRY. An interactive effect of biofertilizer and fertilization was also observed in GBRY.

**Table 3.** Yield components for feed rice (*Oryza sativa* L. cv. 'Fukuhibiki' and line LTAT-29).

| Variety | Distance | Fertilization | Biofertilizer | NSP | PRR | WBR | TPN | GBRY |
|---|---|---|---|---|---|---|---|---|
| | | | | Panicle$^{-1}$ | % | g | m$^{-2}$ | kg |
| 'Fukuhibiki' | 15 × 30 | N2 | Control | 109 | 87 | 22.8 | 235 | 483 |
| 'Fukuhibiki' | 15 × 30 | N2 | BF | 115 | 90 | 23.1 | 282 ** | 650 ** |
| | | | (BF/Control) | (1.06) | (1.03) | (1.01) | (1.20) | (1.34) |
| 'Fukuhibiki' | 15 × 30 | N4 | Control | 114 | 89 | 23.2 | 247 | 581 |
| 'Fukuhibiki' | 15 × 30 | N4 | BF | 102 | 91 | 23.2 | 313 ** | 651 |
| | | | (BF/Control) | (0.89) | (1.03) | (1.00) | (1.27) | (1.12) |
| 'Fukuhibiki' | 30 × 30 | N2 | Control | 129 | 85 | 23.0 | 179 | 429 |
| 'Fukuhibiki' | 30 × 30 | N2 | BF | 108 * | 79 | 23.5 | 210 ** | 471 |
| | | | (BF/Control) | (0.84) | (0.94) | (1.02) | (1.17) | (1.10) |
| 'Fukuhibiki' | 30 × 30 | N4 | Control | 131 | 79 | 22.7 | 200 | 417 |
| 'Fukuhibiki' | 30 × 30 | N4 | BF | 117 | 89 | 24.0 * | 212 * | 485 |
| | | | (BF/Control) | (0.89) | (1.13) | (1.06) | (1.06) | (1.16) |
| LTAT-29 | 15 × 30 | N2 | Control | 215 | 71 | 18.8 | 158 | 438 |
| LTAT-29 | 15 × 30 | N2 | BF | 212 | 72 | 19.2 | 181 ** | 507 * |
| | | | (BF/Control) | (0.99) | (1.01) | (1.02) | (1.15) | (1.16) |
| LTAT-29 | 15 × 30 | N4 | Control | 220 | 64 | 18.6 | 180 | 454 |
| LTAT-29 | 15 × 30 | N4 | BF | 222 | 63 | 18.8 | 189 | 453 |
| | | | (BF/Control) | (1.01) | (0.98) | (1.01) | (1.05) | (1.00) |
| LTAT-29 | 30 × 30 | N2 | Control | 248 | 67 | 17.9 | 128 | 369 |
| LTAT-29 | 30 × 30 | N2 | BF | 268 | 67 | 18.1 | 146 * | 390 |
| | | | (BF/Control) | (1.08) | (0.99) | (1.01) | (1.14) | (1.06) |
| LTAT-29 | 30 × 30 | N4 | Control | 272 | 63 | 17.9 * | 146 | 360 ** |
| LTAT-29 | 30 × 30 | N4 | BF | 274 | 59 | 16.0 | 149 | 279 |
| | | | (BF/Control) | (1.01) | (0.93) | (0.89) | (1.02) | (0.70) |
| ANOVA (*p* value) | | | | | | | | |
| Biofertilizer | | | | n.s. | n.s. | n.s. | <0.000 | <0.000 |
| Variety | | | | <0.000 | <0.000 | < 0.000 | <0.000 | <0.000 |
| Distance | | | | <0.000 | 0.001 | 0.01 | <0.000 | <0.000 |
| Fertilization | | | | n.s. | n.s. | n.s. | <0.000 | n.s. |
| Biofertilizer × Variety | | | | n.s. | n.s. | n.s. | <0.000 | <0.000 |
| Biofertilizer × Distance | | | | n.s. | n.s. | n.s. | 0.002 | 0.005 |
| Biofertilizer × Fertilization | | | | n.s. | n.s. | n.s. | n.s. | 0.007 |

NSP: Number of spikelets in a panicle, PRR: Percentage of ripened rice seeds to the total number of seeds, WBR: Weight of one thousand gross brown rice, TPN: Total panicle number in square meters, GBRY: Gross brown rice yield in 100 square meters (kg); BF: Biofertilization treatment. n.s.: No significance. * and ** indicate significant differences at $p \leq 0.05$ and 0.01 levels between control and biofertilizer application in each treatment, respectively. $n = 3$.

Scatter plots of the correlations of the yield components are shown in Figure 4. In LTAT-29, the NSP with and without inoculation were significantly correlated, whereas no such correlation was observed in 'Fukuhibiki'. The LTAT-29 were plotted on the area of over 1:1 and the slope of the approximation straight line was over 1.0 (Table S8), indicating that the biofertilizer positively affected NSP. The PRR of inoculated and uninoculated LTAT-29 was also correlated with the slope values over 1.0, while 'Fukuhibiki' showed no such correlation, which means the biofertilizer affected positively in high PRR values in LTAT-29 while it did not in low PRR values (Table 3). Finally, the TPN of inoculated and uninoculated 'Fukuhibiki' were significantly correlated and plotted on the area of over 1:1, while that of LTAT-29 was not correlated.

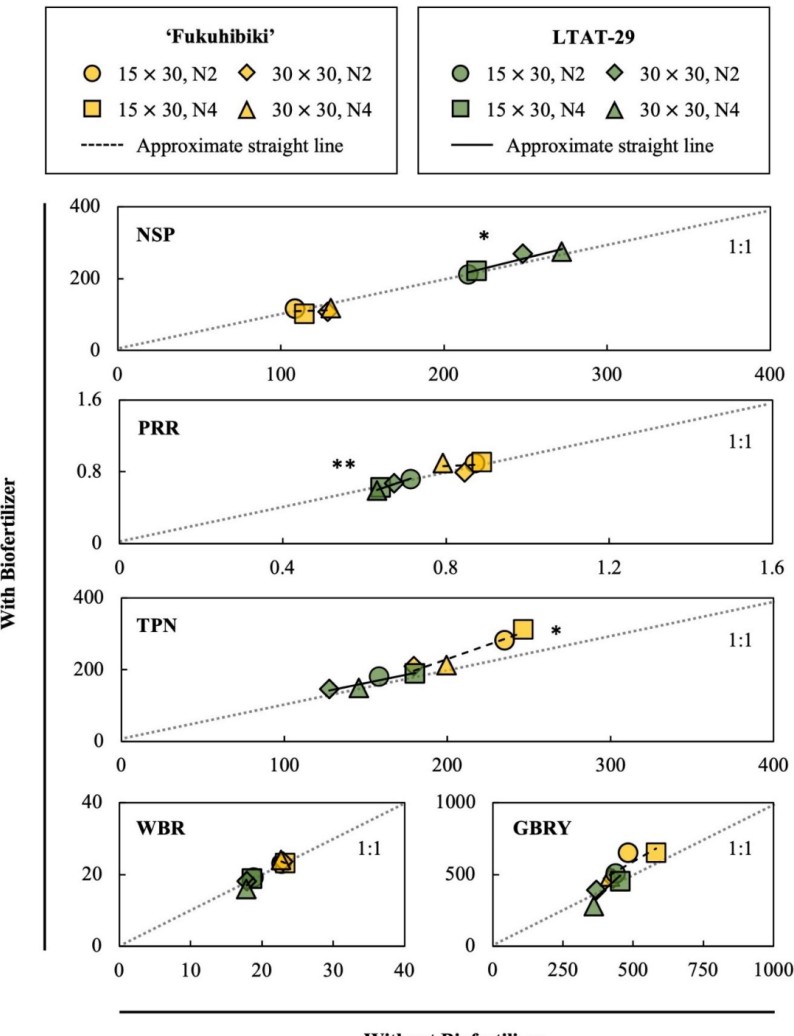

**Figure 4.** Scatter plots of the yield components of feed rice cv. 'Fukuhibiki' and line LTAT-29. The vertical bar of each scatter plot box indicates the value of the inoculated biofertilizer. The horizontal bar of each scatter plot box indicates the values for the non-inoculated plants. The yellow box plots show the values of 'Fukuhibiki', and the approximately straight lines of 'Fukuhibiki' are indicated by dotted lines. The plots of the green circles show the values of LTAT-29, and the approximate straight lines of LTAT-29 are indicated by black lines. Gray dotted lines 1:1 correspondence. NSP: Number of spikelets in a panicle, PRR: Percentage of ripened rice seeds to the total number of seeds, WBR: Weight of one thousand gross brown rice, TPN: Total panicle number in a square meter, GBRY: Gross brown rice yield in 100 square meters (kg). * and ** indicate significant correlation at $p \leq 0.05$ and 0.01 levels, respectively (Pearson's test, two-sided).

### 3.5. The Effects of the Biofertilizer on the Fodder Rice Yields

The analyzed WCS yield components and quality of TAT-26 are shown in Table 4 and Table S9, respectively. The biofertilizer treatment affected the dry weight of straw, which is an important factor for obtaining high biomass in WCS rice. In particular, BF significantly increased the dry weight of straws under 30 × 30 cm and N4 treatments. In contrast to BF, transplanting distance affected other components such as TPN, dry weight of panicles, dry weight of total biomass, the ratio of panicle biomass to total shoot biomass, the nitrogen concentration of the shoot, nitrogen uptake of the shoot, and YTDN. The dry weight of straws, panicles, panicle to total biomass ratio, and nitrogen uptake of the shoots were affected by the nitrogen fertilization treatment. The biofertilizer also improved the nitrogen uptake of the shoots under 30 × 30 cm and N4 treatments. Interaction effects were not observed in the yield components of TAT-26. In terms of WCS quality, CFA and Oa were significantly reduced by BF. At the conventional distance (i.e., 15 × 30) with N4, OCC was increased by BF while CFA and OCW were decreased.

**Table 4.** Yield components for whole crop silage rice (*Oryza sativa* L. line TAT-26).

| Distance | Fertilization | Biofertilizer | TPN | Straw DW | Panicle DW | Total DW | Panicle/Total | N Conc. | N Uptake | TDN | YTDN |
|---|---|---|---|---|---|---|---|---|---|---|---|
| | | | $m^{-2}$ | $g\ m^{-2}$ | $g\ m^{-2}$ | $g\ m^{-2}$ | | $mg\ g^{-1}$ | $g\ m^{-2}$ | $mg\ g^{-1}$ | $g\ m^{-2}$ |
| 15 × 30 | N2 | Control | 113 | 613 | 500 | 1112 | 0.45 | 7.48 | 8.27 | 630 | 700.4 |
| 15 × 30 | N2 | BF | 120 | 653 | 503 | 1156 | 0.43 | 7.36 | 8.52 | 631 | 729.1 |
| | | (BF/Control) | (1.06) | (1.07) | (1.01) | (1.04) | (0.97) | (0.98) | (1.03) | (1.00) | (1.04) |
| 15 × 30 | N4 | Control | 128 | 653 | 458 | 1111 | 0.41 | 7.68 | 8.55 | 628 | 697.6 |
| 15 × 30 | N4 | BF | 123 | 714 | 446 | 1160 | 0.38 | 8.15 | 9.42 | 628 | 729.2 |
| | | (BF/Control) | (0.96) | (1.09) | (0.97) | (1.04) | (0.93) | (1.06) | (1.10) | (1.00) | (1.05) |
| 30 × 30 | N2 | Control | 96 | 610 | 394 | 1003 | 0.39 | 7.36 | 7.37 | 628 | 630.3 |
| 30 × 30 | N2 | BF | 97 | 641 | 394 | 1035 | 0.38 | 7.27 | 7.54 | 628 | 649.6 |
| | | (BF/Control) | (1.01) | (1.05) | (1.00) | (1.03) | (0.97) | (0.99) | (1.02) | (1.00) | (1.03) |
| 30 × 30 | N4 | Control | 97 | 664 | 322 | 985 | 0.33 | 8.05 | 7.92 | 628 | 618.7 |
| 30 × 30 | N4 | BF | 103 | 742 ** | 298 | 1040 | 0.29 | 8.37 | 8.69 * | 627 | 652.8 |
| | | (BF/Control) | (1.06) | (1.12) | (0.93) | (1.06) | (0.87) | (1.04) | (1.10) | (1.00) | (1.06) |
| ANOVA (*p* value) | | | | | | | | | | | |
| Biofertilizer | | | n.s. | 0.037 | n.s. | n.s. | n.s. | n.s. | n.s. | n.s. | n.s. |
| Distance | | | <0.000 | n.s. | <0.000 | 0.007 | <0.000 | 0.006 | 0.045 | n.s. | 0.007 |
| Fertilization | | | n.s. | 0.015 | 0.014 | n.s. | <0.000 | n.s. | 0.026 | n.s. | n.s. |
| Biofertilizer × Distance | | | n.s. | n.s. | n.s. | n.s. | n.s. | n.s. | n.s. | n.s. | n.s. |
| Biofertilizer × Fertilization | | | n.s. | n.s. | n.s. | n.s. | n.s. | n.s. | n.s. | n.s. | n.s. |

TPN: Total panicle number in square meters, DW: Dry weight, TDN: Total digestible nutrients, YTDN: Yield of total digestible nutrients, BF: Biofertilization treatment. n.s.: No significance. * and ** indicate significant differences at $p \leq 0.05$ and 0.01 levels between control and biofertilizer application in each treatment, respectively (two-sided). *n* = 3.

### 3.6. The Effects of the Biofertilizer on the Lodging Resistance of Forage Rice

The biofertilizer significantly affected all analyzed factors, including plant height, fresh weight of shoots, culm numbers, pushing resistance per hill, pushing resistance per culm, lodging moment, and lodging index (Table S10). The numbers in the heatmap displaying the ratio of biofertilizer inoculated to non-inoculated related to lodging factors were indicated to understand the results more clearly (Figure 5). A high value of pushing resistance indicates tolerance to lodging. A high bending moment denotes an increase in the force of rice to collapse itself. The decrease in the lodging index indicates improved resistance to lodging. There was no significant reduction in pushing resistance per hill or bending moment due to biofertilizer inoculation. In 'Fukuhibiki', BF reduced the lodging index in the 15 × 30 cm and N2 treatments. The lodging index ratio of BF subjected to non-inoculated in all treatments in LTAT-29 showed significant differences. In LTAT-29, BF increased the lodging index under N2 and decreased the lodging resistance under N4 treatment with 15 × 30 cm or 30 × 30 cm transplanting distances. Biofertilizer inoculation increased the bending moment in LTAT-29 with 15 × 30 cm and N2 treatments, and decreased the pushing resistance per culm in LTAT-29 with 30 × 30 cm and N2 treatments. In TAT-26, the lodging index was significantly reduced in N2 with 15 × 30 cm spacing.

The interaction effect of biofertilizer and distance on the pushing resistance per culm and lodging index is shown in Table S10.

| Variety | Distance | Fertilization | Pushing resistance per hill | Pushing resistance per culm | Bending moment | Lodging index |
|---|---|---|---|---|---|---|
| 'Fukuhibiki' | 15 × 30 | N2 | 1.21 | **1.49** | 1.03 | **0.77** |
| | | N4 | 1.06 | 0.92 | 0.98 | 0.98 |
| | 30 × 30 | N2 | **1.39** | **1.23** | **1.45** | 1.06 |
| | | N4 | 0.96 | 1.07 | 0.99 | 1.14 |
| LTAT-29 | 15 × 30 | N2 | 1.15 | 0.96 | **1.45** | **1.30** |
| | | N4 | **1.59** | **1.42** | 1.07 | **0.66** |
| | 30 × 30 | N2 | 1.01 | **0.81** | 1.19 | **1.23** |
| | | N4 | **1.39** | **1.24** | 1.01 | **0.73** |
| TAT-26 | 15 × 30 | N2 | **1.40** | **1.36** | 1.02 | **0.67** |
| | | N4 | **1.29** | 1.10 | **1.20** | 0.95 |
| | 30 × 30 | N2 | 1.16 | 0.92 | 1.06 | 0.93 |
| | | N4 | 1.01 | 0.95 | **1.17** | 1.28 |

Increase without significance     Decrease without significance

Increase significantly ( ≤ 1.20 )     Decrease significantly ( ≥ 0.80 )

Increase significantly ( ≤ 1.30 )     Decrease significantly ( ≥ 0.70 )

Increase significantly ( > 1.30 )     Decrease significantly ( < 0.70 )

**Figure 5.** Heatmap for lodging factors showing the ratio of biofertilizer inoculated to non-inoculated. Values indicate the ratio of biofertilized/non-inoculated plants. 'Fukuhibiki', LTAT-29, and TAT-26 measured at 119, 137, and 147 days after transplanting, respectively. Lodging degrees measurement were as follows, 'Fukuhibiki' = 0.0, LTAT-29 = 1.0, TAT-26 = 3.0. Bold indicate significant differences at $p \leq 0.05$ and 0.01 levels between control and biofertilizer application in each treatment, respectively (*t*-test, two-sided). $n = 10$.

## 4. Discussion

### 4.1. The Short-Term and Long-Term Effects of the Biofertilizer on the Forage Rice Growths and Yields

Generally, biofertilizers have been developed with $10^6$–$10^9$ CFU g$^{-1}$ of PGPBs in carriers [16,44,53,54]. In our study, the three rice varieties responded to $10^5$ and $10^7$ CFU mL$^{-1}$ of *B. pumilus* TUAT1 (Table S3), whereas $10^5$ did not solely show the enhancement of each variety (Figure 1). Therefore, $10^7$ CFU g$^{-1}$ is the recommended concentration of this strain for biofertilizer preparation. This strain is already reported that improves plant root growth, and some recorded plant growth-promoting mechanisms on the genome of this strain are known as concentration dependence [10,22]. IAA could inhibit plant growth when it is high concentration [55]. A higher concentration of acetoin also does not increase plant growth [20]. The ideal bacterial concentration for an inoculant might be related to those mechanisms, although the bacteria colonized on plants metabolize one by one differently. However, this result supports that our previous report that was also conducted by ideal concentration [56].

The effects of the biofertilizer containing *B. pumilus* TUAT1 were not the same as those of the resuspended culture inoculation, although the same concentrations of $10^7$ CFU mL$^{-1}$ and $10^7$ CFU g$^{-1}$ were utilized (Figures 1 and 2, Tables S3 and S4). This might be caused by differences in the growth environment of the rice. The nursery box was very shallow and

narrow, providing lesser space for root growth compared with the plant box in our experiment; the height of the nursery box and plant box were 28 mm and 102 mm, respectively. Moreover, four seeds of rice were grown on 300 g of soil in the plant box (75 g seed$^{-1}$), while 3000 seeds grew on 4 kg of soil in the nursery box (1.33 g seed$^{-1}$); therefore, the density of seeds in the nursery box soil was 56 times higher than that in the plant box. These are two of the reasons why *B. pumilus* TUAT1 did not enhance root growth in the nursery box, while the resuspended culture did in the plant box. However, Win et al. [40] reported the promotion of root biomass and no enhancement of shoot biomass in the rice cultivar 'Koshihikari' with biofertilizer containing *B. pumilus* TUAT1, grown in a nursery box, under experimental conditions (size of the nursery box, weight of soil and seeds, harvested period, and shoot length) similar to those in our experiment. The biofertilizer used by Win et al. [40] was a prototype formulated granules of mixed zeolite and silica. The timing of the release of *B. pumilus* TUAT1 from the biofertilizer was controlled because the spores were stored in the small pores of the zeolite carrier. Varietal differences in rice, the type of carrier, and the initial inoculant concentration are also important factors in determining the different effects of *B. pumilus* TUAT1 on roots or shoots.

In the field, biofertilizer inoculation did not improve plant height in the early vegetative growth stage of rice (Figure 3b and Table S6), even though the inoculated plants of all varieties were taller than the uninoculated plants before transplanting (Figure 2 and Table S4). The tiller number at the early vegetative growth stage was not affected by *B. pumilus* TUAT1 colonization in the roots at transplanting; meanwhile, the SPAD value was improved (Figure 3a,c, Tables S5 and S6). *Bacillus pumilus* TUAT1 has been reported to enhance root development, resulting in the promotion of nutrient absorption [22,40]. These results and reports suggest that the improved SPAD values at the early and late vegetative growth stages of rice were derived from enhanced root development promoted by *B. pumilus* TUAT1 after transplanting into the field. Therefore, the increase in tiller number and plant height during the late vegetative growth stage might also be due to improved nitrogen uptake. However, TAT-26 did not show any improvement, suggesting that there is a varietal difference in the responses to the biofertilizer in the vegetative growth stage of rice. During the early and late vegetative growth stages of rice in the field, the biofertilization constantly had interactive effects with transplanting distance on tillers number and plant height, with fertilization on height, and with variety on SPAD values.

In the yield components of feed rice 'Fukuhibiki' and LTAT-29 (Table 3), the statistical improvement in the GBRY was a result of the increase in the TPN due to biofertilizer inoculation. However, a treatment of biofertilizer with 30 × 30 cm and N4 significantly reduced GBRY in LTAT-29 due to the lower weight of rice seeds (i.e., the weight of one thousand gross brown rice). The brown rice yields of each variety with biofertilization, 15 × 30 cm (i.e., conventional distance), and N2 (i.e., basal fertilization) were the highest and was significantly improved due to the enhanced root development promoted by the biofertilizer in early growth stages when only basal fertility was required in the field. The yields at N2 with biofertilizer were 12% higher than those of N4 without biofertilizer on both 'Fukuhibiki' and LTAT-29, respectively. This indicates the possibility that biofertilization reduces more than 50% of nitrogen. This result is more remarkable in comparison with a report by Adesemoye et al. [57], where the plant growth-promoting *Bacillus* spp. complemented 25% of chemical fertilizer in tomato plants. On the other hand, the benefit of basal fertilizer to the effect of biofertilizer in LTAT-29 was reported by Win et al. [42]. However, in their study using 12 times higher basal nitrogen fertilizer (i.e., 24 g m$^{-2}$) than that in our treatment, improvement in tiller number due to biofertilizer inoculation was not detected. Adesemoye et al. also observed no improvement in nitrogen uptake of shoot when the PGPB was treated on plants with sufficient fertilization [57]. From our results and those reports, we suggest that rice responds better to an abundance of chemical fertilizer than the inoculation of *B. pumilus* TUAT1, showing sufficient growth that makes it hard to see the effect of biofertilization. In order to express the highest effects, a minimum amount of nitrogen is desirable, leading to reduced chemical fertilizer usage in the world of

sustainable agriculture. In addition, the sparse transplanting distance significantly reduced the percentage of ripened rice seeds, lightened the seeds, and suppressed the yield. In theory, the rice plants in 30 × 30 cm spacing have access to twice the amount of nutrients than those in the 15 × 30 cm spacing because the density of plants in the field is 50% less. Generally, release-controlled fertilization contributes to seed ripening and formation [58,59]. The timing of additional fertilization during cultivation also affects seed formation. For example, ripening is delayed by fertilization at the heading stage and PRR is reduced by fertilization at the early vegetative growth stage [60,61]. Based on these results, sparse transplanting decreased the PRR in our experiment, resulting in the suppression of GBRY. Biofertilization interacted with transplanting distance on the GBRY, which is ascribed to the interaction effects between them on TNP (Table 3). The scatter plots (Figure 4) indicate that NSP and PRR of LTAT-29 (panicle weight-type rice) and TPN of 'Fukuhibiki' (panicle number-type rice) was increased in response to *B. pumilus* TUAT1. From the results, we propose that the increased yields by biofertilizer inoculation were due to the increased yield components, and the response of the yield components to inoculation depended on the rice variety.

### 4.2. The Contribution of the Biofertilizer to the Quality of Fodder Rice as WCS and Lodging Resistance Improvements of Forage Rice

In the yield components of WCS rice (Table 4) TAT-26, improved nitrogen uptake at 30 × 30 cm and N4 due to biofertilizer inoculation significantly increased the straw dry weight and not the TPN, panicle dry weight, TDN, and YTDN, indicating that the biofertilizer only affected the dry weight of the straw. It has been reported that rice's varietal and seasonal differences are critical to the effectiveness of PGPBs and biofertilizers in improving plant growth, feed yields, and biomass yields [53,62]. Therefore, TAT-26 might be a variety that responds well to the biofertilizer under the abundance of chemical fertilizer contrary to the feed rice (i.e., 'Fukuhibiki' and LTAT-29). The sparse transplanting distance reduced the TPN, dry weight of panicles, and total biomass per square meter because the number of rice plants in the plots of 30 × 30 cm was half of that at 15 × 30 cm. However, the dry weight of straw was not significantly different with the differences of transplant spacing, indicating that the amount of straw from one rice plant in sparse transplanting was approximately twice that obtained from a plant in the conventional distance. The TDN was not significantly improved by biofertilizer inoculation because the decreased Oa offset the effect, and the OCC was increased (Table S9). Kusa et al. [63] reported that the ratio of panicle biomass to the total shoot biomass of the cultivar 'Leaf Star' decreased with the amount of nitrogen fertilizer. Nitrogen fertilization in our experiment also reduced the panicle to total shoot biomass ratio because it increased the dry weight of straw and significantly decreased the dry weight of panicles. The yields of total biomass and digestible nutrients were significantly higher at the conventional transplanting distance of 15 × 30 cm (Table 4), suggesting that the density of rice plants in the field was the most important factor for high yields in our treatments. Interactive effects of the biofertilizer with transplanting distance or fertilization were not observed in the present study.

As for the effects of biofertilizer on lodging resistance, the lodging index of LTAT-29 under N2 with conventional transplanting distance was increased because the bending moment was significantly increased (Figure 5 and Table S10); the plant height and fresh weight were increased. The increasing lodging index in sparse transplanting can be ascribed to the decrease in pushing resistance per culm. These results indicate that biofertilizer inoculation increases the lodging index through various mechanisms. In contrast, all decreases in the lodging index due to biofertilizer inoculation were related to pushing resistance per culm. *Bacillus pumilus* TUAT1 improves nutrient uptake by enhancing root growth [22,40]. Our field survey results at the late growth stage of LTAT-29 showed that the SPAD values of plants with N4 and both conventional and sparse distances had improved (Figure 3f). Therefore, the roots of inoculated rice plants might have developed more than those of the uninoculated rice plants, resulting in higher resistance to pushing,

which is a direct contribution of biofertilizer to higher lodging resistance and nitrogen concentration. Another factor for the indirect contribution was the culm strength, which was probably promoted by the improved photosynthesis due to higher nitrogen contents in the leaf at the late vegetative growth stage and by the creation of carbohydrates such as fibers. Zhang et al. [64] also reported that the culm diameters of two japonica rice varieties were improved by nitrogen fertilization, supporting our hypothesis. However, the concentration of carbohydrate fibers (e.g., ADF and OCW) in TAT-26 under improved conditions, i.e., $15 \times 30$ cm and N2 treatments, was not enhanced (Table S9). Hence, we propose that the mechanism of lodging index reduction by biofertilizer inoculation could be as follows: the improved root development (which is particularly important in the direct sowing cultivation of rice) due to biofertilizer inoculation increased the resistance to pushing and the uptake of nitrogen to synthesize carbohydrates, creating tougher culms, resulting in higher resistance to lodging. However, varietal differences in the effectiveness of biofertilizers also appeared in the lodging index. The interactive effect of biofertilizer with the transplanting distance was shown on pushing resistance per culm, resulting in those on the lodging index (Table S10).

## 5. Conclusions

The biofertilizer containing *Bacillus pumilus* TUAT1 spores at $10^7$ CFU $g^{-1}$ improved the yields as feed rice and fodder rice in the paddy field. It also changed the quality of fodder rice as WCS and the lodging resistances. These results are desirable for further application of this biofertilizer for sustainable agriculture practices, such as reducing the amount of nitrogen fertilizer used. Our study also indicated that the differences in variety, fertilization, and transplanting density affected the effectiveness of *B. pumilus* TUAT1 on the vegetative growth stages, yields, and lodging resistance in the field. Here, the greatest issue in the utilization of biofertilizers may be the different varietal effects in rice. Hence, breeding new rice varieties that respond positively to biofertilizer application and developing new biofertilizers that do not show any selectivity for rice varieties are required for sustainable agriculture in the future.

**Supplementary Materials:** The following supporting information can be downloaded at: https://www.mdpi.com/article/10.3390/agronomy12102325/s1, Table S1. Analyzed soil characters of the paddy filed. Table S2. Results for ANOVA on the resuspended culture inoculation test ($n = 3$). Table S3. Results for Dunnett's test (two-sided) on the resuspended culture inoculation test ($n = 3$). Table S4. Summary of the seedlings inoculated with Kikuichi/Yume-bio on the nursery box at transplanting stage. 'Fukuhibiki' was harvested at 21 days after sowing. LTAT-29 and TAT-26 were harvested at 14 days after sowing. BF: Biofertilizer, RSWL: Ratio of shoot fresh weight to the shoot length. * and ** indicate significant differences at $p \leq 0.05$ and 0.01 levels between control and biofertilizer application in each treatment, respectively. Standard deviation (SD) was shown (*t*-test, two-sided). $n = 3$. Table S5. Colonized numbers of *B. pumilus* TUAT1 at basement of the root at the transplanting period. CFU: Colony forming unit. Table S6. Summary plant growth surveyed after 8 weeks transplanting. TN: Tillers number. BF: Biofertilizer. n.s.: No significance. * and ** indicate significant differences at $p \leq 0.05$ and 0.01 levels between control and biofertilizer application in each treatment, respectively (*t*-test, two-sided). $n = 3$. Table S7. Summary plant growth surveyed after 13 weeks transplanting. TN: Tillers number. BF: Biofertilizer. n.s.: No significance. * and ** indicate significant differences at $p \leq 0.05$ and 0.01 levels between control and biofertilizer application in each treatment, respectively (*t*-test, two-sided). $n = 3$. Table S8. Summary of the slope of the approximation straight lines, regression analysis ($R^2$) and Pearson correlation test (two-sided) for yield components of 'Fukuhibiki' and LTAT-29. NSP: Number of spikelets in a panicle, PRR: Percentage of ripening rice seeds to total numbers, WBR: weights of one thousand gross brown rice, TPN: Total panicle numbers in a square meter, GBRY: Gross brown rice yield in 100 square meters. F: 'Fukuhibiki', L: LTAT-29. Table S9. The quality as WCS, line TAT-26. The unit of each value is milligram per gram (mg $g^{-1}$). CP: Crude protein, CFA: Crude fat, CFI: Crude fiber, CA: Crude ash, NFE: Nitrogen free extract, NFC: Non-fiber carbohydrate, ADF: Acid detergent fiber, NDF: Neutral detergent fiber, OCC: Organic cellular content, OCW: Organic cell wall, Oa: Organic a fraction in OCW (hydrolyzed rapidly by

cellulase), Ob: Organic b fraction in OCW (resistant to cellulase digestion). BF: Biofertilizer. n.s.: No significance. * and ** indicate significant differences at $p \leq 0.05$ and 0.01 levels between control and biofertilizer application in each treatment, respectively (*t*-test, two-sided). $n = 3$. Table S10. Summary of effects of BF inoculation on the lodging factors of rice. FW: Fresh weight. BF: Biofertilizer. n.s.: No significance. * and ** indicate significant differences at $p \leq 0.05$ and 0.01 levels between control and biofertilizer application in each treatment, respectively (*t*-test, two-sided). $n = 10$. Figure S1. Circumstances surrounding animal forage in Japan. (a) Area of the forage rice cultivation (feeds rice and fodder rice) according to the reports of the Ministry of Agriculture, Forestry and Fishers (MAFF), and the importing price of animal forage according to the Trade Statistics of Japan. The prices were converted to USD by the calendar based yearly average exchange rates reported by BOJ Time-Series Data Search. (b) Domestic self-sufficiency rate of forage reported by MAFF. Figure S2. Pedigree chart of the analyzed new forage rice lines (LTAT-29 and TAT-26). Figure S3. Seedlings on the plant box. Cont.: Control (non-inoculated).

**Author Contributions:** S.-i.A.: Conceptualization, Methodology, Data curation, Formal analysis, Investigation, Visualization, Writing—Original draft preparation. Y.O.: Conceptualization, Methodology, Investigation, Resource. K.K.: Methodology, Investigation. E.Y.: Methodology, Investigation. M.D.A.R.: Methodology, Writing—Reviewing and Editing. S.D.B.-K.: Writing—Reviewing and Editing. T.Y. (Tetsuya Yamada): Writing—Reviewing and Editing. T.O.: Conceptualization, Writing—Reviewing and Editing, Resource. N.O.-O.: Supervision, Writing—Reviewing and Editing. T.Y. (Tadashi Yokoyama): Supervision, Conceptualization, Writing—Reviewing and Editing, Project Administration, Funding acquisition. All authors have read and agreed to the published version of the manuscript.

**Funding:** This research was supported by a grant from the Project of the NARO Bio-oriented Technology Research Advancement Institution: The Special Scheme Project on Developing Regional Strategy (Grant No. 16822446).

**Data Availability Statement:** All relevant data can be accessible within the paper and Supporting Information files.

**Acknowledgments:** WISE Program; Doctoral Program for World-leading Innovative & Smart Education of TUAT supported by the Ministry of Education, Culture, Sports, Science and Technology (MEXT), Japan. The authors thank Ohno's Farm (Tatsuhiro Ohno, Miwako Ohno, and Mizuki Ohno), a member of Yuukinosato Towa Organic Village in Nihonmatsu city, Fukushima prefecture and Koji Matsukawa at Tokyo University of Agriculture and Technology for their kind cooperation and sincere help with field experiment and management.

**Conflicts of Interest:** This research was conducted with a research fund from NARO which one co-author belongs to.

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
