# Peer review of "Biofertilizer with Bacillus pumilus TUAT1 Spores Improves Growth, Productivity, and Lodging Resistance in Forage Rice"

_agronomy, doi:10.3390/agronomy12102325_

Round 1

Reviewer 1 Report

The introduction is very comprehensive and to the point. Methodology sufficient. Did the authors refer to any standards? Results: Please argment the selection of the statistical test and the number of repetitions. The discussion was very extensive and sufficient. Concise and factual conclusions. Please check the edit Please check the English language by a native speaker

Reviewer 2 Report

Dear authors, the subject of the manuscript is very interesting. I think this type of study should be encouraged. However, the discussion is not up to the "idea", it needs to be improved, talking about the mechanisms by which this type of "fertilization" increases the growth variables. Why do microorganisms act like this? What are the interactions with plants and soil, how can this interaction affect plant growth? How it can affect photosynthesis, how it can improve nutrient absorption.

Therefore, I recommend that an opportunity be given, as the topic is relevant, but the discussion should be made agin.

Best regards

Reviewer 3 Report

The major points are:

1-    English language should be revise well

2-    Put all latin names in italics throughout the text (it is wrong in quite many places, check them one by one); put also the names of the author(s) of all taxa cited the first time they appear in the text

3-    Keyword should be different than in title

4-what is the source of the bacteria 

5-how long you added the seed in sodium hypochlorite

6-what the condition of greenhouse? 

7-some time you wrote days after planting and some time you wrote week please write the same

8- Make sure that all scientific names in the References list are italics.

9- if you will write DOI for paper please add the DOI for ALL the References

10- Fig no 1 not clear

11- All tables must be self-explanatory.

Round 2

Reviewer 2 Report

Dear authors,

I would like to see an improvement in the layout of images and an improvement in the discussion. I don't particularly consider comparing results with other research a discussion. Unless the mechanism by which they are equal or not is explained. They could put more relative, current references. And I still think they could discuss a little more in the sense of work innovation.

For example: what are the advantages of your treatments? Isn't it the fact that mineral sources are limited? How can your treatments enhance nutrition? How can microorganisms act, is the energy expenditure large in this assimilation? Can there be savings in chemical fertilization for this crop? What would the potential savings be?
